# Rehabilitation outcomes of bird-building collision victims in the Northeastern United States

Ar Kornreich[1], Dustin Partridge[2]*, Mason Youngblood[3,4], Kaitlyn Parkins[2,5]

1 Fordham University Graduate School of Arts and Sciences, Bronx, New York, United States of America, 2 NYC Bird Alliance, Inc, New York, New York, United States of America, 3 Max Planck Institute for Geoanthropology, Minds and Traditions Research Group, DE, Jena, Germany, 4 Institute for Advanced Computational Science, Stony Brook University, Brook, New York, United States of America, 5 American Bird Conservancy, The Plains, Virginia, United States of America

* dpartridge@nycbirdalliance.org

**Data Availability Statement:** Github: (https://anonymous.4open.science/r/AvianWindowCollisions-1FF9) Zenodo (https://doi.org/10.5281/zenodo.8280339).

## Abstract

**Building** collisions are a leading threat to wild birds; however, only those that are found dead or fatally wounded are included in current mortality estimates, with injured or stunned birds largely assumed to survive long-term. Avian building collision victims are often brought to wildlife rehabilitators for care, with the hopes they can be released and resume their natural lives. We examined the wildlife rehabilitation records of over 3,100 building collisions with 152 different avian species collected across multiple seasons to identify patterns of survival and release among patients. The number of admissions varied by season; fall migration had the highest number of cases and winter had the least number of cases, and summer having the lowest release proportion and winter having the highest. The most common reported injury was head trauma and concussion. Our logistic and Poisson models found that mass had a strong positive effect on release probability, and the season of summer had a strong negative effect on release probability. Mass and winter had a strong positive effect on treatment time, and age and the seasons of fall and winter had a strong negative effect on treatment time in these models. Ultimately, about 60% of patients died in care, either by succumbing to their injuries or by euthanasia. Patients that were released remained in care for longer than patients that died. This study reports different data than carcass studies and views bird-building collisions from the perspective of surviving victims to explore longer-term effects of these collisions on mortality. Increased communication and collaboration between wildlife rehabilitators and conservation researchers is recommended to better understand building collisions and how to respond to this leading threat to wild birds. These findings, along with our estimate of delayed mortality, suggest that overall collision mortality estimates based on carcass collection far exceed one billion birds in the U.S. each year.

## Introduction

Even common avian species in North America are declining [1], so identifying and addressing threats to bird populations is critical for the conservation of not only individual species, but

**Funding:** The author(s) received no specific funding for this work.

**Competing interests:** The authors have declared that no competing interests exist.

the ecosystems they belong to [2]. Collisions with buildings are a leading anthropogenic cause of death for birds in the United States [3], with major collision events garnering national attention [4] and, as of 2024, 25 U.S. cities or states have passed legislation that requires new buildings to be bird-friendly [5]. Affecting over 50 avian families and hundreds of species [6]; building collisions kill between 365 million and 1 billion birds a year in the United States alone [7] and pose a significant threat to birds in other countries as well [8–10].

Deadly collisions occur at both multi-story buildings and individual homes and in both urban and rural areas [7, 11]. In North America collisions during fall migration typically surpass the other seasons [12–15]; however, frequent collisions are documented in the spring, summer, and winter months as well [13,16–19]. A number of factors endanger birds attempting to navigate in areas with buildings, many of these stemming from the difference in perception between humans and birds. Glass is transparent and largely invisible, so whereas humans are taught the concept of glass early in development and identify visual cues to identify it (but will still occasionally collide with it), birds do not understand glass and do not know or learn these visual cues [20]. Presumably, birds perceive the image reflected off of glass or seen through glass as potential habitat or open space, especially open sky, or vegetation [21]. Greenspace near reflective windows can be problematic as a result, and studies of bird-building collisions regularly consider the presence, height, and distance of vegetation from glass as risk factors [6,22–26]. Architectural factors like glass area [23,24,27,28], building size [29] and individual building facade characteristics [30] have been shown to influence collision numbers. For homes, bird feeders have also been associated with increased collisions [31–33].

Artificial light at night (ALAN) further exacerbates the risk of avian collisions with buildings, particularly for migratory species [34]. At broad scales, ALAN attracts nocturnal migrants, causing them to concentrate in urban areas [35,36] where they are in proximity to a high density of potentially deadly glass buildings. At the finer scale, ALAN emitted from individual buildings is associated with increased collisions at those locations [37–39]. Weather conditions can also play a role in risk of collisions for nocturnal migrants by affecting the density of migrants that stopover in a given location, the behavior of these migrants, or by amplifying the effects of ALAN [14,15].

Most studies examining building collisions are conducted on recovered carcasses of collision victims that die on impact [6,7,15,40]. However, though over 80% of collisions with buildings were once estimated to be immediately fatal to birds [23], more recent studies have not supported this pattern and suggest that far fewer than 80% of collision victims die on site. For instance, in Korner et al.'s 2022 study [39], only 7.2% of casualties were found dead. Therefore, estimates based on bodies found at the scene of collisions might provide a drastic underestimate of the true toll of building strikes on bird populations.

Some birds survive the immediate collision and move away from the scene [41]. Possibly, the survivors may recover from their sustained injuries, but they may be more vulnerable to predation or succumb to these injuries with time [42–45] or suffer reduced fitness and reproductive disadvantages [46]. Though post-collision prognosis for a disoriented or injured bird may be bleak, building collision survivors that are brought to wildlife rehabilitators are thought to have the best chance of recovery and subsequent survival. To fully understand the crisis of building collisions, these delayed deaths, or potentially disadvantaged survivors, matter, especially if these survivors are less of an exception to the rule than previously thought.

Knowing how many birds hit buildings, survive, and fly away without intervention is impossible, let alone what becomes of them later. The closest cases we have available to observe what happens to birds that do not die upon impact are those taken to wildlife rehabilitators. The International Wildlife Rehabilitation Council (IWRC) defines wildlife rehabilitation as "the treatment and temporary care of injured, diseased, and displaced indigenous animals, and

the subsequent release of healthy animals to appropriate habitats in the wild" [47,48]. Wildlife rehabilitators, licensed by their state, can either practice as private individuals or as part of an organization, usually non-profit [49,50]. Due to the nature of their work, wildlife rehabilitators often find themselves mediating human-wildlife conflict or mitigating the effects of anthropogenic threats to wildlife [48,51–53]. Wildlife rehabilitators, owing to their unique vantage point, may have knowledge that is evident among themselves, but may be less apparent or known to others working in other conservation fields, as databases of wildlife rehabilitation and rescue information, despite having great potential to inform the understanding of many conservation issues in the face of anthropogenic threats, remain underutilized by researchers [49,54]. While wildlife rehabilitation data has been used to track pathogens and poisonings [53,55–57], or identify potential threats of concern toward wildlife and particularly vulnerable species [11,58–60], these studies represent only a small fraction of how rehabilitation data can inform conservation research and policy [51,53].

This study aims to provide an example of wildlife rehabilitation data as applied to a pressing conservation issue. Wildlife rehabilitators have long been educators of the public about wildlife and mediators in human-wildlife conflict, providing a unique front-line vantage point for many crises that face wildlife. This study is an effort to merge the knowledge and perspective of rehabilitation outcomes with collision research and advocacy.

The first goal is to examine patterns of survival, identifying which factors, such as age, mass, season of admission, and organizational funding, affect a patient's time in treatment and ultimately, the outcome of their case (whether they are released, presumably recovered, or deceased, by euthanasia or dying on their own, usually referred to as the patient's disposition). The factors that affect a patient's outcome may be different from the factors that affect a patient's initial survival of the collision. Examining these factors may provide insight into how these collisions are happening and how collisions may be affecting different bird species.

The second goal is to investigate avian building collisions from a new perspective, and to highlight that carcass surveys almost certainly provide an underestimate of collision casualties. Birds, even if they survive the initial impact with the glass, may not truly survive the ordeal but rather die later from their injuries. This delayed death may remain even when removed to a safe environment and medically treated, which theoretically improves their chance at survival. Examining delayed deaths even under the most ideal of conditions can greatly inform our understanding of avian collisions with building structures and windows.

## Methods

### Acquiring rehabilitation patient data

We submitted Freedom of Information Act/Law requests (or the state equivalent) to eight states in the northeast and mid-Atlantic (Connecticut (CT), Delaware (DE), Massachusetts (MA), Maryland (MD), New Jersey (NJ), New York (NY), Pennsylvania (PA), and Rhode Island (RI)) requesting records regarding avian building collision cases (cases) between 2016 and 2021. We requested information including species, dates received, disposition (outcome), disposition date, and location.

Most license-issuing states require the submission of annual records of rehabilitation activity, but how much and what kind of information collected in annual records varies by state (Table 1). Some states request very minimal information, often only requiring tally forms, such as in New Jersey, Connecticut, and Massachusetts.

Furthermore, the format of rehabilitation data varied widely in level of detail and completion where specific forms were not mandatory for collecting information, such as in Pennsylvania. There was further heterogeneity in the level of detail contained even with standardized

**Table 1. Information required by state law in wildlife rehabilitator annual reports By US States.**

| Information/State | NY | PA | RI | DE | D.C | MA | NJ | CT |
|---|---|---|---|---|---|---|---|---|
| **Species** | Yes | Yes | Yes | Yes | Yes | Yes | Yes | Yes (2) |
| **Disposition (outcome)** | Yes | Yes | Yes | Yes | Yes | Yes | Yes | Yes |
| **Date of Admission** | Yes | Yes | Yes | No | Yes | No | Yes | No |
| **Date of Disposition** | Yes | Yes | No | No | Yes | No | Yes | No |
| **Cause of Admission** | Yes (1). | Yes | Yes (3) | No | Yes | No | No | Yes (2) |
| **Injury information** | No | Yes | Yes (3) | No | Yes | No | No | No |
| **Age of Patient** | Yes | Yes | Yes | Yes | No | No | No | No |

Information required by state law in wildlife rehabilitator annual reports in seven United States states and one federal district of the United States. These requirements hold for all wildlife rehabilitation patients, not just birds. (1) New York has an extensive system of standardized codes to differentiate and organize causes of admission that rehabilitators are required to use in their reports. (2) Connecticut forms contain highly limited choices regarding species and causes of admission; songbirds are all lumped together into one row on the tally sheet, and causes of admission are limited to "Hit By Car", "Attacked by Dogs," or "Attacked by Cats." (3) Cause of Admission and Injury information are often found lumped together, guaranteeing the presence of information for neither; for instance, Cause of Admission may contain simply "broken wing", which only provides injury information, or "hit by car", which gives no information on sustained injuries.

forms as fields can be left blank and circumstances of admission or injuries left vague, and rehabilitators varied in what kind of information records contain even with databases shared with other rehabilitators [49].

Cases were only included if the patient was a bird (Class: Aves), admitted due to a collision with a building (including window, glass storefront, door) or strong suspicion thereof (e.g., "suspected window strike," "probably hit window"), that was admitted between 2016 and 2021. Cases were omitted if the cause of admission was perhaps not a building or window strike, such as "found on ground" or "maybe hit by car". Cases also were omitted if the disposition was listed as "transferred" or "pending", as their final disposition was not known. There was only one case with the disposition "kept for education". Additionally, injury data from Tri-State Bird Rescue and Research was omitted due to the complex nature of reporting and coding incompatibility.

Due to variation in language used to describe injuries, injuries were categorized into more general categories (for instance, humerus and ulnar fractures were all classed as "wing fracture", dull mentation and depressed affect was classed as "lethargy"), except for injuries whose language was extremely consistent ("Concussion", "Head trauma", "Unconscious", "Unable to fly", "Skull Fracture"). Raw data and categorization rules are included in the Supplementary Materials accessible through the links in the Data and Code Availability Statement.

Cases were classified as major city (population density 5,000 people per square miles or more), urban (between 4,999 and 1,000 people per square mile), suburban (between 999 and 100 people per square mile), and rural (less than 99 people per square mile), based on county of location. Population density information for counties was provided by USA.com's [61] population density rankings and the 2010 US Census. To examine the effect of organizational funding we obtained the average annual income listed for each organization from ProPublica's version of IRS form-990 filings, averaged across the years for which that organization contributed to the dataset.

Species mass data and family information were taken from Cornell Lab of Ornithology All About Birds website, averaged high and low values. Definitions of seasons were based on migration dates [7]. March, April and May, when migrant species are traveling northward to their summer ranges, was defined as spring; June and July, when most birds are breeding and rearing young, was defined as summer; August, September, October, and November, when

migrant species are traveling south to their winter ranges, was defined as fall; and December, January, and February, when migrant birds are residing in their southern ranges and resident birds are primarily focused on foraging, was defined as winter.

## Missing data imputation

There were several variables for which we have incomplete data: sex (90%), age (43%), organization type (14%), funding (17%), and treatment time (17%). Sex was missing far too much of the data and was excluded from analysis. For everything else, we handled missing data using the multiple imputation approach recommended by Bürkner [62], a method that has been shown to produce more accurate estimates than both single imputation (e.g., replacing NA with averages) [63] and complete case analysis (e.g., dropping incomplete rows from the data) [64,65]. First, we used multiple imputation with random forest machine learning to generate 10 different imputed versions of our data using the mice package in R [66]. All predictors and outcome variables included in the statistical modeling (treatment time, release, death, mass, funding, season, and age; see below) were also included during multiple imputation. Then, we fit our models to each imputed dataset separately and pooled the results across all of the models by simply combining the posterior distributions for each parameter [62].

## Statistical modeling

All models were Bayesian and were run in STAN using the brms package in R [62], fit with 20,000 iterations across four MCMC chains. Categorical variables were converted to factors and continuous variables were scaled and centered for efficient model convergence. All analysis code, prior specifications, and model diagnostics can be found in the Supplementary Materials accessible through the Data and Code Availability Statement.

Major drivers of rehabilitation were identified using two methods: (1) logistic and Poisson models of release and treatment time, and (2) using a right-censored survival model of treatment time where the censored cases are those where birds died and could not be released. The logistic and Poisson families were used because release and treatment time are binary and count variables, respectively. Logistic models are simply binomial models with a logit link-function. A right-censored survival model is a survival model where the survival times are missing for some individuals, due to death or other factors. We only included mass, funding, season, and age as fixed effects in the modeling. Causal analysis with a directed acyclic graph suggested that family, species, state, and type of organization were unnecessary for estimating effects (see Supplementary Materials accessible through the Data and Code Availability Statement). The final specification of the first two models was as follows, with all predictors as fixed effects:

$$release \sim mass + age + funding + season$$

$$treatment \sim mass + age + funding + season$$

For the survival analysis, we used the parametric approach described by Kurz [67] and inspired by McElreath [68] using the brms package in R [62]. We modeled survival as a Weibull distribution, as a simple Weibull model outcompeted the other two common parametric forms: exponential and Gamma (see Supplementary Materials accessible through the Data and Code Availability Statement). Our survival analysis was right-censored, meaning that treatment time was the outcome variable, and the censored cases were those where birds died and

could not be released. The specification for the survival model was as follows:

$$\text{treatment} \mid \text{censored(death)} \sim \text{mass} + \text{age} + \text{funding} + \text{season}$$

It is important to note that our data violated the assumption of non-informative censoring, or the assumption that the censored variable (death) is independent of the event variable (release) [69]. There are imputation methods that replace the event times of censored cases to account for bias introduced by non-informative censoring, but they are designed for cases where full data are available, or where missing data imputation is not also in use [70]. In this study we did not use specialized imputation for informative censoring, so the overall release curves will be systematically biased towards shorter times. We did not have a reason to expect this to bias predictor estimates, but the results of the logistic and Poisson models should be given greater attention than the survival model.

Descriptive statistics, Pearson Chi-Square Test, Fisher's Exact Test, Wilcoxon and Kruskal-Wallis rank sum tests, and Kolmogorov-Smirnov tests were calculated in R (version 4.2.2) and its built-in functionality.

## Results

### Sources of case data

We received responses from six states (NJ, CT, MA, NY, PA, and RI), two never responded (MD and DE), though, the Tri-State Bird Rescue and Research rehabilitation organization, based in DE, provided their building-collision cases independently from their digital database (Wildlife Rehabilitation Medical Database, also known as WRMD). Three of the states that responded could not provide any records matching our requests (NJ, CT, and MA). One state (NY) was only able to provide records from 2020–2021, from selected counties (we selected Kings, Queens, New York, Nassau, Suffolk, Bronx, Westchester, Erie, Monroe, and Albany counties based on proximity to major cities and human population). Two states (PA and RI) produced all relevant records between 2016 and 2021, and one of these (RI) had fully digitized the data and provided a searchable spreadsheet; the other (PA) had simply sent all rehabilitation records from all species for all admission reasons for the time frame requested as non-indexed portable document format (.pdf) files (as did NY), and relevant cases were manually identified and transcribed into the dataset.

The dataset included a total of 3,159 included rehabilitation cases, with 1,644 cases from PA, 564 from DE, 418 from NY, 408 from RI, and 35 cases from other states brought to a rehabilitation center in the states examined (27 from NJ, 2 from CT, and 6 from Washington, D. C.). Of 3,159 cases, 74 could not be traced to the county level, 140 cases were from rural counties, 1,461 cases were from suburban counties, 1,387 were from urban counties, and 97 were from counties defined by major cities, with 62 cases from Philadelphia County PA, 13 cases from New York County NY, 6 cases each from Washington D.C. and Queens County NY, 5 cases from Kings County NY, 3 cases from Bronx County NY, and 2 cases from Baltimore County MD. New York County was vastly underrepresented in our dataset because data from the largest wild bird rehabilitator in New York City could not be obtained for analysis.

Our dataset contained 152 different avian species and 42 different avian families. Mourning Doves (*Zenaida macroura*) (n = 258), American Robins (*Turdus migratorius*) (n = 188), Northern Cardinals (*Cardinalis cardinalis*) (n = 187), Cooper's Hawks (*Accipiter cooperii*) (n = 174), and Gray Catbirds (*Dumetella carolinensis*) (n = 171) were the most common species patients. Picidae, Columbidae, Turdidae, Accipitridae, and Cardinalidae were the most common families in the patient dataset. Fig 1 plots the average mass of species in each family

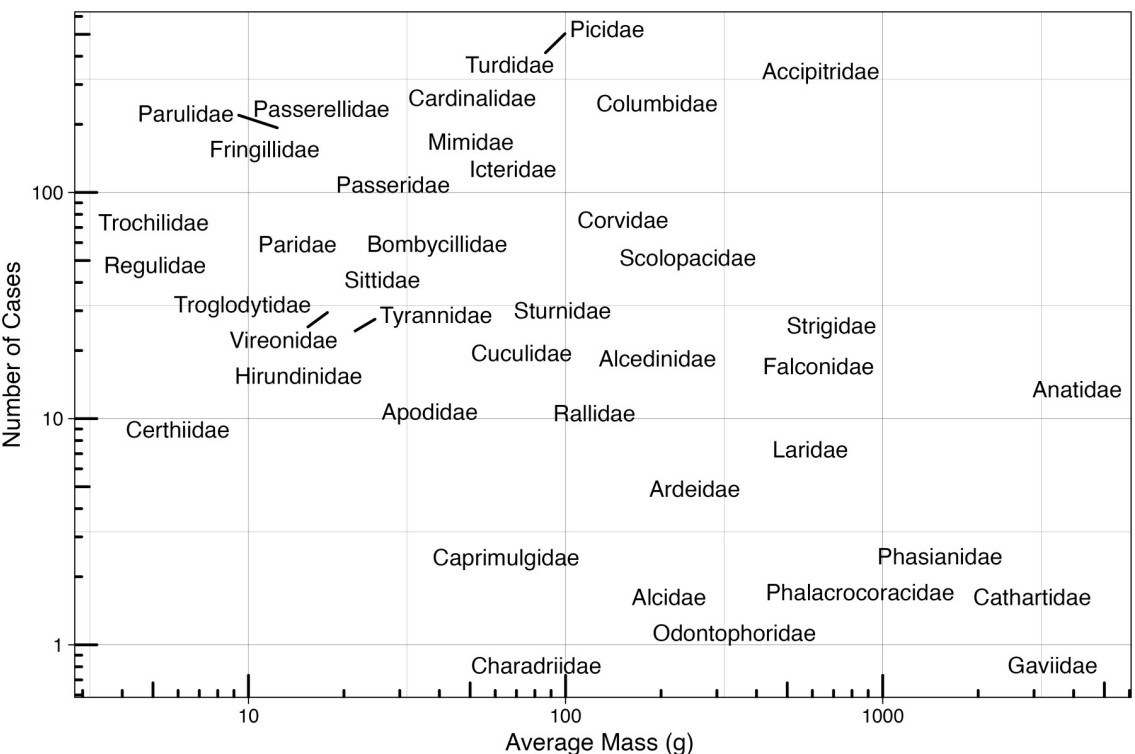

**Fig 1. Avian families, plotted by mass, of wildlife rehabilitation cases stemming from avian-building collisions with known outcomes.** Families were plotted based on the average mass of the species of that family represented in the building collision rehabilitation cases, and their number of cases in which they appeared in the rehabilitation records relating to building collisions.

and relative abundance of cases in each family and Table 2 reports all families with more than 20 cases, and their release statistics.

## Case outcomes and treatment times

Of these 3,159 cases, 126 were dead on arrival. From the remaining 3,033 cases, 39.5% resulted in release (n = 1197), 32.1% resulted in unassisted death during treatment (n = 974), 28.4% resulted in euthanasia (n = 861), and 1 case resulted in the patient being kept as an educational animal ($<$0.1%).

Of the 3,159 cases, 127 could not be used for further analysis because birds were dead on arrival or 3,032 cases that omit dead-on-arrival cases and the kept-for-education, resulting in 3,032 cases available for further study. Patients that were released averaged 12.89 days in treatment (median: 7 days), with 27.15% of releasable patients released within the first 2 days of treatment, and 51.21% of releasable patients released within the first 10 days of treatment. Patients that died or were euthanized averaged only 2.64 days in treatment (median: 1 day), with 33.79% dying on the same day as intake, and 57.44% dying within the first 2 days of treatment. Difference in treatment time between released and deceased patients was significant ($p < 2.2e\text{-}16$, Wilcoxon and Kruskal-Wallis rank sum test; W = 344901 and Kruskal-Wallis chi-squared = 546.11, df = 1; $p < 2.2e\text{-}16$, Kolmogorov-Smirnov test, D = 0.4684).

Admission of building-collision patients varied by season; 579 cases were admitted in the spring (March through May, 21.6%), 626 cases admitted in the summer (June through July, 23.3%), 1,063 cases were admitted in the fall (August through November, 39.6%), and 417

**Table 2. Release and total wildlife rehabilitation cases stemming from avian-building collisions for families containing more than 21 patients, with release rates.**

| Family | Released | Total | Release Rate | Family | Released | Total | Release Rate |
|---|---|---|---|---|---|---|---|
| Strigidae | 19 | 30 | 0.6333 | Cardinalidae | 81 | 214 | 0.3785 |
| Falconidae | 13 | 21 | 0.619 | Mimidae | 63 | 192 | 0.3281 |
| Parulidae | 92 | 181 | 0.5083 | Regulidae | 12 | 38 | 0.3158 |
| Vireonidae | 14 | 28 | 0.5 | Trochilidae | 18 | 58 | 0.3103 |
| Accipitridae | 131 | 270 | 0.4852 | Passeridae* | 38 | 139 | 0.2734 |
| Bombycillidae | 19 | 41 | 0.4634 | Troglodytidae | 9 | 33 | 0.2727 |
| Passerellidae | 80 | 174 | 0.4598 | Cuculidae | 6 | 24 | 0.25 |
| Turdidae | 129 | 286 | 0.451 | Corvidae | 22 | 90 | 0.2444 |
| Columbidae | 132 | 295 | 0.4474 | Icteridae | 25 | 103 | 0.2427 |
| Fringillidae | 59 | 137 | 0.4307 | Scolopacidae | 10 | 42 | 0.2381 |
| Sittidae | 14 | 33 | 0.4242 | Tyrannidae | 5 | 24 | 0.2083 |
| Picidae | 157 | 395 | 0.3975 | Sturnidae* | 5 | 36 | 0.1389 |
| Paridae | 17 | 43 | 0.3953 | | | | |

Families are in the order of percentage released. Number of cases per family ranged from 395 to 1. Note that the smaller the number of cases, the more subject to chance release rate is, and considered with more caution.

\* Families containing only invasive species.

cases were admitted in the winter (December through February, 23.3%). In years with more than 100 cases (2016–2021, but not 2014 or 2015), fall always had the most cases and winter always had the least, but summer cases outnumbered spring in 2017, 2019, and 2020.

The proportion of patients that were released after intake also varied by season, with summer having the lowest release proportion (0.29), and winter having the highest release proportion (0.46), leaving spring and fall with comparable proportions (0.39 and 0.40, respectively). Unlike admissions, in years with more than 100 cases (2016–2021), no season consistently had the highest or lowest release rate, and no season's release rate was always higher or lower than another. We found a significant effect of season on final disposition (p = 1.08e-8 < 0.001 for Pearson Chi-Square Test, and Fisher's Exact Test, for Pearson's Chi Square, $\chi^2$ = 39.973, df = 3; p = 7.64e-9 for Fisher's Exact Test.).

A total of 1,267 cases were missing age information (42%), 1366 cases were adult birds (45.1%), and 399 cases were juvenile birds (13.2%). The proportion of adults that were released was larger than that of juveniles (605 adult releases, 44.3%; 154 juvenile releases, 38.6%). The effect of age on final disposition was significant (p = 0.0496 < 0.05 for Pearson Chi-Square Test, $\chi^2$ = 3.855, df = 1; p = 0.0444 for Fisher's Exact Test).

## Injury statistics

The dataset we analyzed contained 1,217 cases with information regarding injuries patients were suffering from. The most common injuries reported were head trauma (534 cases, 34.8% released), concussion (257 cases, 66.9% released), balance issues (172 cases, 24.1% released), wing fractures (110 cases, 18.2% released), and inability to fly (106 cases, 49.1% released). Injuries with the highest release rates were concussions (66.9%, 172 releases of 257 cases), shock or stunned (61.4%, 35 released of 57 cases), hypothermia (50.0%, 1 released out of 2 cases), inability to fly (49.1%, 52 released out of 106 cases), and bruising or swelling (43.7%, 38 released out of 87 cases). No other injury type had more than a 35% release rate. No cases reporting paralysis (15 cases), unconscious patient (4 cases), or skull fractures (3 cases), resulted in release. Injuries with the lowest release rates were lethargic or weak patients (7.4%, 2 released out of 27

**Table 3. Results of the probabilistic models on release and treatment time of avian patients in wildlife rehabilitation care after building collisions.**

| Release Model (Logistic) | | | | | Treatment Time Model (Poisson) | | | | |
|---|---|---|---|---|---|---|---|---|---|
| **Variables** | **Est.** | **2.5%** | **97.5%** | **Sig Eff.** | **Variables** | **Est.** | **2.5%** | **97.5%** | **Sig. Eff.** |
| Mass | 0.14 | 0.05 | 0.24 | * | Mass | 0.09 | 0.08 | 0.10 | * |
| Age | 0.13 | -0.07 | 0.34 | | Age | -0.20 | -0.31 | -0.09 | * |
| Funding | -0.04 | -0.12 | 0.04 | | Funding | -0.01 | -0.04 | 0.03 | |
| S: Summer | -0.47 | -0.70 | -0.23 | * | S: Summer | -0.30 | -0.42 | -0.19 | * |
| S: Fall | 0.02 | -0.18 | 0.22 | | S: Fall | -0.12 | -0.18 | -0.05 | * |
| S: Winter | 0.24 | -0.01 | 0.48 | | S: Winter | 0.30 | 0.24 | 0.38 | * |

2.5% and 97.5% mark the lower and upper bounds of the 95% credible intervals. Season conditions, with spring as the reference condition, are marked with a preceding "S:". The asterisks denote significant effects: 95% credible intervals that do not overlap zero.

cases), leg fractures (11.1%, 3 released out of 27 cases), back, spine, or neck injuries (13.2%, 9 released out of 69 cases), nerve damage (13.6%, 3 released out of 22 cases), and internal bleeding (15.4%, 2 released out of 13 cases).

The vast majority of patients appeared otherwise healthy apart from injuries sustained from collision. Of the 1217 cases with reported injury data, only 33 patients were emaciated or dehydrated (2.7%), and 29 had infections, parasites, or were otherwise sickened (2.4%). The dataset also included 6 cases with animal attacks in connection with the collision (usually after collision had occurred but before the patient reached rehabilitation); only one such patient was released of the six.

## Modeling release and treatment time probabilities

The results for both the logistic model of release and Poisson model of treatment time are shown in Table 3 and Fig 2. The logistic model showed only mass and the season of summer to have a strong effect on release probability, with mass having a positive effect on release probability (95% CI: 0.05, 0.24), and summer having a negative effect on release probability (95% CI: -0.70, -0.23). This suggests that species with greater mass had better odds of release, and patients admitted in the summer had poorer odds of release than those admitted in the spring. No other season (fall or winter, relative to spring), nor organizational funding, had a

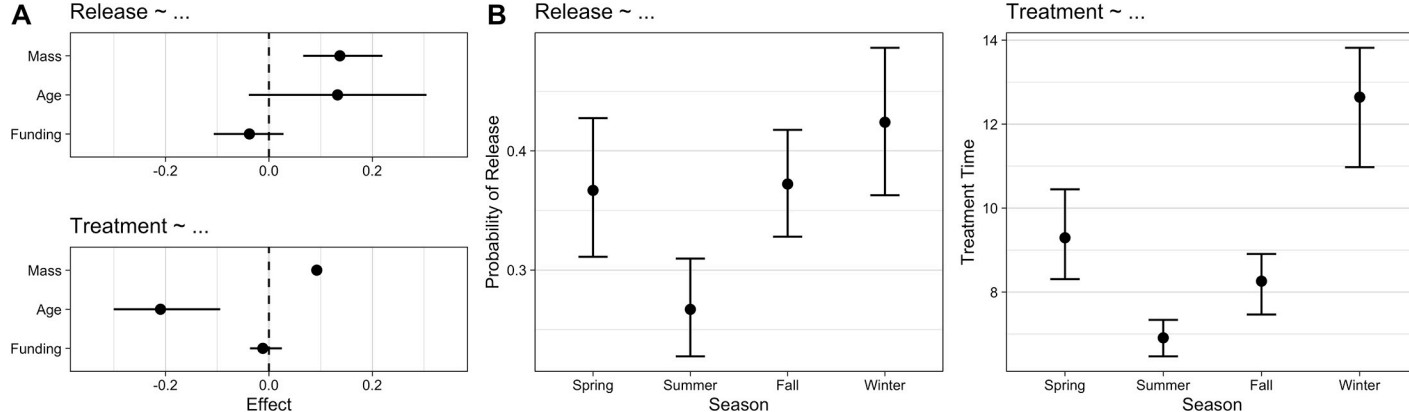

**Fig 2. Visualized results of release and treatment time models for rehabilitated avian building collision victims.** Results of the logistic model of release and the Poisson model of treatment for avian patients in wildlife rehabilitation care after building collisions. Panel A shows the effects of mass, age, and funding, whereas panel B shows the marginal effects of season. In panel A, the lines correspond to 95% credible intervals. Intervals that do not overlap zero denote significant effects.

**Table 4. Results from survival model analysis of treatment time for rehabilitated avian building collision victims.**

|  | Estimate | Est. Error | 2.5% CI | 97.5% CI | Sig. Effect |
|---|---|---|---|---|---|
| Intercept | 3.21 | 0.11 | 2.99 | 3.42 |  |
| **Mass** | 0.13 | 0.04 | **0.06** | **0.21** | * |
| **Age (Adult)** | -0.24 | 0.10 | **-0.44** | **-0.05** | * |
| Funding | -0.00 | 0.03 | -0.06 | 0.06 |  |
| Season: Summer | 0.05 | 0.11 | -0.16 | 0.27 |  |
| Season: Fall | -0.13 | 0.08 | -0.28 | 0.03 |  |
| Season: Winter | 0.13 | 0.10 | -0.06 | 0.32 |  |

Results of the Weibull survival model for avian patients in wildlife rehabilitation care after building collisions. 2.5% and 97.5% mark the lower and upper bounds of the 95% credible intervals. The asterisks denote significant effects: 95% credible intervals that do not overlap zero.

significant effect on release probability. The effect of age trends is strongly positive, although the wide credible interval still overlaps with zero (see Table 3). The Poisson model showed strong positive effects of mass (95% CI: 0.08, 0.10) and the season of winter (95% CI: 0.24, 0.38) on longer treatment time, and strong negative effects of age (95% CI: -0.31, -0.09) and the seasons summer (95% CI: -0.42, -0.19) and fall (95% CI: -0.18, -0.05) on treatment time. There was still no effect of funding on treatment time (95% CI: -0.04, 0.03).

### Survival modeling and analyses

The Weibull survival model found that mass had a strong positive effect on treatment time (95% CI: 0.06, 0.21), and adult age had a strong negative effect on treatment time (i.e., decreases treatment time, 95% CI: -0.44, -0.05, Table 4). These results agree with the Poisson model of treatment time, that treatment time increases with mass and decreases with age. However, there was no effect of season or organizational funding on treatment time (95% CIs overlapped 0), although fall trends negative relative to spring. Posterior predictions from the Weibull model can be found in the Supplementary Materials accessible through the Data and Code Availability Statement.

## Discussion

### Building collision survival rates

Our findings demonstrate that the number of deaths attributed to collisions is significantly underestimated and that well over 1 billion birds are dying each year in the United States. We found that even with active treatment and care from rehabilitators, 60% of building-collision patients die of their injuries or are euthanized. Our findings are especially concerning when considering that death rates in the wild may be even higher than 60% if collision victims are not found and treated by rehabilitators. Loss et al. included only definitive mortalities to estimate the number of dead birds killed in the U.S. each year, but injured birds with unknown outcomes equaled 14.9 percent of the number of dead birds found during the collision monitoring the study used [7]. If 60% of those birds die from their injuries, the upper boundary of the total estimate is likely at least 10% too low, and could be increased to 1.092 billion birds every year; however recent monitoring efforts using 24-hour collision patrols and video monitoring suggest that estimates based on carcass surveys far underestimate the number of birds that collide with buildings [39,41], and hence the number of birds that die but leave no evidence behind. A case study on a building's light pollution in Germany found most birds attracted to and colliding with the building were found alive but disoriented enough to be

caught by hand before being released; only 7% were found already dead on site [39]. Another recent study using video monitoring found that nearly 80% of birds moved off-site directly following a collision and only 7% of birds died immediately following impact [41]. These findings, along with our estimate of delayed mortality, suggest that overall collision mortality estimates based on carcass collection far exceed one billion birds in the U.S. each year. Delayed mortality contributes to underestimation of collision deaths and should be considered alongside carcass removal and predation when adjusting for underestimation [64]. While collisions that do not stun birds may be less impactful to wellbeing and kill fewer birds (i.e., ricocheting birds caught on film in Samuels et al. [41]), birds that are stunned enough to be captured by observers are unlikely to survive. Many released patients seem to require multiple days to recover to releasable condition, a luxury that is not offered to most birds outside of rehabilitation.

## Factors affecting survival

When collision victims are brought to rehabilitators, cases resulting in release had longer treatment times than cases resulting in death. While this difference may partially stem from the practice of euthanasia, standard wildlife rehabilitation guidelines and practices call for euthanasia once a patient is determined to be unreleasable, rather than prolong an unviable or captive life [47,71]. However, case information reveals that many collision victims take at least a day in treatment before death, highlighting the sometimes-gradual lethality of collision incidents even when victims do not die at the scene. On the other hand, patients that survived and were released often presented a greater time investment for wildlife rehabilitators, particularly with juvenile patients who may have needed extra care during this stage of their development. Treatment time may also vary based on the severity or type of injuries incurred (e.g., fractures) or the protocols and practices of individual rehabilitators or facilities (e.g., not releasing in poor weather conditions).

The seasonal abundance of building collision cases follows the previously described pattern of being highest in fall [7,12], but departs from this pattern by having more cases overall in the summer than spring (although see [27]). In the northeast there are significant changes between seasons that affect bird presence and behavior. Winter, the coldest months, generally occur from Mid-November through Mid-March. March through May coincides with spring migration, when birds are moving northward to breeding grounds. The breeding season, when most birds nest and breed, is June through Mid-August. From Mid-August through Mid-November is fall migration, when most birds are moving southbound for the winter [7]. However, this difference may be due to changes in human behavior between the seasons that would allow for more observation of collisions or injured birds in the summer, when more people in the northeast United States tend to be outside, than during spring migration, and thus more birds brought to rehabilitators. However, upon admission, summer had the highest rate of patients dying during care and had a negative impact on release probability (although fall had the most deaths overall). This decreased release rate may be due to the increased proportion of juvenile patients [72], whose release rates are poorer than adult patients. This effect could also be due to differences in avian behavior during the summer, as for most species breeding occurs during that time [72] and breeding territorial behavior may contribute to collisions as they react to perceived territorial threats apparent in window reflections. Potentially, a difference in species composition with an increased prevalence of species with poorer release rates may explain this variation at least partially, though no particular family or species with migratory behavior was found to explain this variation; despite the abundance of Gray Catbirds (*Dumetella carolinensis*) in the dataset, and this species' vulnerability to building-collision deaths [7], their

removal did not change this effect. Additionally, though not statistically significant, winter had the highest release rate of the seasons. A decreased prevalence of species with low release rates, decreased patient traffic ("baby season" in the spring and summer tend to be the busiest times for wildlife rehabilitation centers [73]), or changes in species behavior during winter foraging [18,74] may account for this difference.

Species mass was positively associated with release probability, suggesting that larger species, once admitted, are more likely to survive building collisions and recover to a releasable state. This result is contrary to the findings of a study of carcasses surveyed by Veltri and Klem [75], finding that larger birds sustained more severe injuries. However, these findings are consistent with other rehabilitation studies [76]. Mass was also positively associated with treatment time. The longer treatment time may reflect increased survival, as cases ending in release had longer treatment times, presumably because they did not die or were euthanized midway through treatment. The increased probability of survival with larger mass may stem from differing collision physics linked to flight speed or other behavioral differences. Biological differences may also account for this effect; owls (Strigidae) in particular can survive certain ocular injuries that would necessitate euthanasia in other species (Anne Lewis, personal communication). Analysis of injury type prevalence and relationship to size or taxonomic family would provide further insight into the mechanism of this effect, perhaps similar to the work conducted on a Toronto-based data set by Hudecki and Finegan [43].

Although the logistic model of release found that age was not a strong predictor of release, the effect of age trends strongly positive in the logistic model and the Fisher's Exact Test and Pearson Chi-Square Test found such an effect to be significant, suggesting that adults had more releases than juveniles. Nevertheless, adult collision victims outnumber juvenile cases for every season, including summer and fall when juvenile cases were most numerous. This was unexpected, given juvenile casualties often outnumber adult casualties in studies [7,72,77,78, though see 79]. The prevalence of adults in rehabilitation cases may be due to a difference in ability to survive the initial collision, and juvenile birds may be more likely to die upon impact [77]. Alternatively, juvenile plumage may render younger victims harder for concerned humans to detect and bring to a rehabilitator. Nevertheless, birds do not escape the risk of collision by remaining in good health, as only slightly more than 5% of patients were suffering from infection, sickness, parasites, dehydration or emaciation. Building collisions are therefore killing off members of the population in a non-selective manner, worsening the populations it affects by removing otherwise fit individuals [80].

Neither probability of release, nor treatment time, were found to be affected by organizational funding of rehabilitation centers or facilities. While well-funded organizations can treat more collision victims, increasing funding does not seem to affect release rate. While release and treatment time were not affected by funding, funding may still be important for bird well-being. Rehabilitation centers are dependent on funding for operations, purchasing new supplies, and supporting hard working staff. The 40% of birds that do survive rehabilitation undoubtedly benefit from care, and funding to support the well-being of these birds remains necessary.

## Conclusions

Communication between the fields of scientific research and wildlife rehabilitation is critical for increased understanding of the building collision crisis, and subsequently how to mitigate its impact on bird populations. However, significant barriers complicate efforts to incorporate the highly important knowledge and perspectives of wildlife rehabilitators. Both rehabilitators themselves and their licensing state can and should take measures to address these issues. Though most states require case logs and/or tallies from rehabilitators, the relevant agencies

do not appear to have developed ways to examine, organize, or keep track of the data these records contain, a missed opportunity to monitor possible wildlife issues in their states. The quality and thoroughness of the records themselves are also quite heterogeneous. Descriptions of injuries varied from thorough to extremely vague (simply "bruising" or "injured"), if they were included at all, with great variability in the phrasing of injuries and reason for intake (i.e. "hit window", "flew into window", "building collision", "hit glass door"). Ideally, a standardized interstate database of all building collision victims would be ideal, but in lieu of that, licensing states can modify required forms and how they are processed, and individual rehabilitators or rehabilitation organizations can collaborate with researchers and each other to establish a database or modify existing databases (such as WRMD and WILD-ONe) to facilitate data-sharing and analysis [60,81–83]. However, wildlife rehabilitation staff and volunteers are often overworked and underfunded, making accurate and detailed record keeping difficult. Increased funding and guidance from state agencies or assistance from conservation researchers could help rehabilitators improve and maintain records that more effectively aid in conservation research and care for patients. To the detriment of birds, wildlife rehabilitators are not typically included in ornithological conferences, and their studies are often restricted to rehabilitation-specific journals not often reviewed by academic conservationists, resulting in limited dissemination of information across disciplines. More cooperation between scientists and rehabilitators may increase the dialogue between the two perspectives, to the benefit of both [84].

While examining rehabilitation patients is perhaps the closest glimpse into what happens to birds after collisions, this model does not offer a perfect view. Bias likely affects which birds are brought to rehabilitators [49,60]. Easily seen birds are preferentially brought in as smaller or more camouflaged species may go unnoticed. Release rate does not represent the chances of surviving a building strike, but rather of recovery if one has already survived and was taken for treatment. Nevertheless, review of rehabilitation records provides a more complete view of factors affecting collisions and subsequent survival. Including more states and regions in this kind of dataset and a more detailed and nuanced treatment of urbanization within counties would further bolster understanding of rehabilitation record patterns [83]. Additionally, more robust analyses of injury data, species flight styles and behavior, and taxonomic relationships of patients could be highly informative in gaining insight into the effects of collisions and factors affecting survival [81–83,85]. Furthermore, examining rehabilitation records over the course of time would provide further insight into the trajectory of the collision crisis and how collisions may be affecting different populations, and human communities, locally, nationally, and potentially globally [11,58,60,85].

The injuries from collisions, once sustained, have only around 40% combined probability of recovery, even with the highest standards of care. Additionally, information is missing about long-term effects of collision post-rehabilitation, such as behavioral changes or other factors that could affect survival and reproduction in the wild. For these reasons, though 60% chance of death is no doubt better than certain death, prevention of building collisions by making glass visible to birds and reducing artificial light at night should remain the top priority reducing the impact of collisions, and wildlife rehabilitators should continue to educate the public and advocate for implementation of safety measures. Buildings can be made safer for birds by using bird-friendly glass or by retrofitting existing glass with applied bird-friendly films. These products reduce collisions using markings that are spaced no more than 2 by 4 inches apart, ideally 2 by 2 inches, and are visible to birds [86].

A realistic understanding of outcomes after collisions is important to exploring and properly investing in second lines of defense. Wildlife rehabilitators, conservation researchers, policy advocates, and citizens are all crucial in the struggle to understand and respond to the calamities wild birds, and wildlife overall, face. The most effective strategy to confront these

realities are through cooperation between the four groups, by sharing knowledge, tools, and perspectives to formulate an informed, balanced response to mitigate a leading anthropogenic cause of death in avian wildlife in the United States.

## Supporting information

**S1 Dataset. Wildlife rehabilitation cases of avian patients admitted due to building collisions.** The complete dataset of all rehabilitation cases included in the analysis are listed in this dataset. "Source Organization" refers to the agency or organization that provided the data, and "Filename/Section" refers to the name of the file and the section of that file each case was found. "Patient Number" was listed with most rehabilitation cases obtained, and was preserved for ease of double-checking transcription. The species codes included are consistent with those used by the Institute of Bird Populations. "RhbFund" reports the average annual income listed for each organization from ProPublica's version of IRS form-990 filings, averaged across the years for which that organization contributed to the dataset.
(XLSX)

**S1 Table. Wildlife rehabilitation building collision injury classifications and descriptions.** Table listing the injury classifications that were used, and notes on injury descriptions and classification rules aiding in binning cases by injury similarity despite highly varied language or word choice.
(PNG)

## Acknowledgments

We thank Dr. J. Alan Clark for advisement in obtaining the data and for mentorship through Fordham University's conservation certificate program. We thank Lisa Smith and Tri-State Bird Rescue and Research for their rehabilitation records. We thank Anne Lewis, Lucas DeGroote, and Tim Jasinski (Lake Erie Nature & Science Center) for information, advice, and suggestions over the course of this project. We thank Scott Loss for contributing advice and ideas toward the updated bird-building collision mortality estimate.

## Author Contributions

**Conceptualization:** Ar Kornreich, Kaitlyn Parkins.

**Data curation:** Ar Kornreich.

**Formal analysis:** Ar Kornreich, Mason Youngblood.

**Investigation:** Ar Kornreich, Kaitlyn Parkins.

**Methodology:** Ar Kornreich, Mason Youngblood, Kaitlyn Parkins.

**Project administration:** Ar Kornreich, Dustin Partridge, Kaitlyn Parkins.

**Software:** Mason Youngblood.

**Supervision:** Dustin Partridge, Kaitlyn Parkins.

**Visualization:** Mason Youngblood.

**Writing – original draft:** Ar Kornreich, Dustin Partridge, Mason Youngblood, Kaitlyn Parkins.

**Writing – review & editing:** Ar Kornreich, Dustin Partridge, Kaitlyn Parkins.

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
