## [Decision Letter · Decision Letter 0]

29 Apr 2024

PONE-D-24-13495Rehabilitation outcomes of bird-building collision victims in the Northeastern United StatesPLOS ONE

Dear Dr. Kornreich,

Thank you for submitting your manuscript to PLOS ONE. After careful consideration, we feel that it has merit but does not fully meet PLOS ONE’s publication criteria as it currently stands. Therefore, we invite you to submit a revised version of the manuscript that addresses the points raised during the review process.

 We received the opinion of four reviewers this week.

Typically, PLOS ONE requires two reviews. This higher number relative to your submission occurred because when I invited potential reviewers, I selected some that have experience with (1) wildlife rehabilitation and survival modeling, and (2) mortality estimates involving birds and window collisions. Due to something out of my control, the first two acceptances occurred for the group 1 of reviewers. Thus, I decided to invite more researchers of group 2, to have a more complete set of opinions. A few days later, two researchers of group 2 agreed to review your submission. Thus, we have four reviews. I ask your patience to consider all of them. It will be an extra work, but together they might substantially improve the quality of your study. I also made my own review, as typically happens, but it is mainly focused on formatting to PLOS ONE.

Reviewer 1 suggested Minor Revision and highlighted the high quality and importance of your study. She/he provided a set of suggestions to improve your study; mostly, they are relative to the methods.

Reviewer 2 suggested Major Revision and provided some compliments on your study. She/he provided a large set of suggestions from the title to the discussion that aim to improve your manuscript.

Reviewer 3 also provided compliments and highlighted the importance of such approach. She/he is mainly concerned with the statistical analysis, and proposed some changes.

Reviewer 4 appreciated your study and suggested Acceptance. She/he made some comments and corrections on the enclosed file.

Please find my own review below. Please submit your revised manuscript by Jun 13 2024 11:59PM. If you will need more time than this to complete your revisions, please reply to this message or contact the journal office at plosone@plos.org. Please include the following items when submitting your revised manuscript:A rebuttal letter that responds to each point raised by the academic editor and reviewer(s). You should upload this letter as a separate file labeled 'Response to Reviewers'.A marked-up copy of your manuscript that highlights changes made to the original version. You should upload this as a separate file labeled 'Revised Manuscript with Track Changes'.An unmarked version of your revised paper without tracked changes. You should upload this as a separate file labeled 'Manuscript'.

We look forward to receiving your revised manuscript.

Kind regards,

Dárius Pukenis Tubelis, Ph.D.

Academic Editor

PLOS ONE

Journal Requirements:

2. Please upload a copy of Supporting Materials which you refer to in your text on pages 7, 8, 9 and 15.

**Additional Editor Comments:**

Please find my review below:

Introduction

Very well written, as occurred with the objectives.

Line 39. Do not indent the first paragraph of a given section. Only the next ones.

Line 46. You have to change to [8 –10]. When you have 3 or more refs in immediate sequence, you have to use a long dash (not hyphen) between the initial and final numbers. Please check this throughout.

Line 48/49. It has to be [11–14].

Line 50. “12” should be the first number within the brackets. Keep always an increasing order.

Line 58. Change to [6, 21–25]. Please follow this format along the manuscript.

Line 71. I think you have to delete “´s 2022 study”. Just keep the author and the brackets with number.

Line 129. You could add birds and US to the table title.

Line 158. Delete the space between the second bracket and the comma. Delete the comma and space after “location”.

Line 165. I think you have to delete “´s 2014 study”.

Line 244. It is better to change to “varied”. Consider always using past in the Results.

Line 262. Fix the brackets in italics.

Lines 264 to 265. As these numbers after families (n = xx) are found in the Table, there is no need to repeat them here. I suggest deleting them from the text.

Line 270. I think that the figure caption has to be more informative (birds, n states, US…).

Line 274. Its title is not much informative. I suggest the same for the figure caption. Why are the names in italics ? (first column and first line of the table).

Line 313. Table 3. Please check if I´m wright or wrong….should you add something like “Variables” in the cell above that of Mass?

Line 331. Table 4. Explain the bold in some cells. Or change it.

Discussion

I suggest you add Sut-titles for sections of the Discussion. It eases the reding and the finding of specific parts. Not mandatory, but very welcome.

Line 255. I think you have to delete “2022”.

Lines 423-425. It is a too short paragraph. Can you merge it with other ?

Lines 413-438. Ops! Here you have a full page (19) without the citation of references/papers. Can you please revise this section, and try to add some studies to compare and or support your thoughts/findings. Ah! Now I saw that this problem extends to line 469. Please try to improve this final part of the Discussion. It is too “clean”. Consider an international audience.

References

Page numbers of articles and chapters have to be separate by a long dash and not hyphen.

You have to add the DOI for all references that have it (mostly articles). Use the format https://doi.org/numbers and letters. Check recent issues if in doubt.

Supplementary Material

Along the text, you used something like “see Supplementary Materials”. You should be more specific, like mentioning the table,  figure, etc… Please check the instructions again. Still about this, I could not find the links for them in the pdf that I have to evaluate your submission. I do not know if you provided them or not during the submission. Please check this when you submit the corrected version ok ? They will be necessary. Maybe it was not your fault but a problem in the system…

Dárius Tubelis

PLOS ONE Editor

Reviewers' comments:

Reviewer's Responses to Questions

**Comments to the Author**

1. Is the manuscript technically sound, and do the data support the conclusions?

Reviewer #1: Yes

Reviewer #2: Partly

Reviewer #3: Yes

Reviewer #4: Yes

2. Has the statistical analysis been performed appropriately and rigorously? 

Reviewer #1: Yes

Reviewer #2: I Don't Know

Reviewer #3: I Don't Know

Reviewer #4: Yes

3. Have the authors made all data underlying the findings in their manuscript fully available?

Reviewer #1: Yes

Reviewer #2: Yes

Reviewer #3: Yes

Reviewer #4: Yes

4. Is the manuscript presented in an intelligible fashion and written in standard English?

Reviewer #1: Yes

Reviewer #2: Yes

Reviewer #3: Yes

Reviewer #4: Yes

5. Review Comments to the Author

Reviewer #1: I consider the results presented in the manuscript to be very beneficial for a better view of the threats to bird species in nature, especially because some of them are still underestimated. The work is clearly written, the literature review and the discussion is of high quality, however, for a better understanding, I suggest some explanation, especially in methodology.

Line 87 - You state that wildlife rehabilitators are licensed by their state - what the conditions are for obtaining a license? I suppose it varies from state to state. Are at least some of the conditions for issuing a license the same or similar, for example, do the conditions for obtaining this license contain certain knowledge (e.g. factors affecting the survival of animals in nature, the basics of treatment, etc.) and their verification, for example by an exam?

Line 244 – Why do some seasons include three months and others only two? I assume that in your latitudes the conditions are different from ours, where the year would be divided into four seasons of three months each. On what basis was the division chosen? Can such a division not affect the results of the number of individuals admitted for the given period?

281 – 292 It is not clear how the individual types of injuries differ from each other - for example, how does a concussion differ from a balance issue, as I assume that even with a concussion, balance problems can occur. I would suggest listing the individual categories in the methodology with an example so that it is clear how they differ from each other. Likewise, information on diagnosis is missing - is a veterinarian present and are any special examinations (for example, X-rays) carried out?

Line 436 - Is it typical for these rehabilitation centres that they care for only one group of animals (e.g. birds) or are all animals admitted to the rehabilitation centres (including mammals, etc.)? This may also reflect the specialization of these centres and thus better knowledge of the given species.

Reviewer #2: Major comments:

In this study, the authors collate a large data set and explore predictors of bird-building collisions throughout the Northeastern United States. The taxonomic scope of the data set is impressive, spanning some 152 spp., and the authors report significant effects of season and body mass, among others, on aspects of bird-building collisions.

Overall, I thought that the study has merit and could be of interest to a wide audience. However, in its current form I feel that it requires major revision before it can be deemed suitable for publication.

Specifically, as a reviewer I often found myself confused. This often indicates that extra clarification is needed to help the reader fully understand what the authors have done and more importantly, why? This is particularly the case surrounding the statistics employed here. There were some statements which I disagreed with, and I have highlighted these in my list of minor comments below. I encourage the authors to read and cite more widely within the wildlife rehabilitation literature – there are many studies that should be acknowledged here.

Importantly, I was left wondering what the a priori expectations of testing age, body mass, funding, etc., would be given previous knowledge on bird-building collisions. At the end of the introduction, I would expect to find the authors’ expectations evidenced by published research. Inclusion of this, and discussion of each point in sequential order in the Discussion would largely improve the paper.

Given the nature of this study and the findings, I also expect the Discussion section to be full of evidence-informed management actions to prevent bird-building collisions in the future. Currently, there are none. What about decals on windows, habitat management, tinting glass, etc.? This seems like an obvious oversight by the authors and I am sure they will welcome the opportunity to apply their findings in such a way.

As previously mentioned, I believe this study has merit and I hope that the authors receive my comments in a constructive manner, in which they were intended.

Please see point-by-point comments below.

Minor comments:

Title/abstract – the title refers to “building collisions” but “window collisions” are used throughout the abstract. I would suggest being consistent in use of terminology.

Abstract

Line 19 – replace second “birds” with “those” (the word birds is included three times in one sentence here).

Lines 22-24 – there seems to be a sentence missing in your summarised methods here. For example, you state that patterns of survival and release were examined but the next sentence begins with seasonal differences. I suggest adding something along the lines of:

“In addition to this, we examined temporal trends in bird-window collisions and compared differences between species across seasons…”.

Lines 24-26 – split into two separate sentences, it will read better.

Lines 26-27 – this seems a bit obvious, consider omitting to save on word count.

Lines 30-31 – this seems to contradict the statement on lines 20-21.

Line 32 – “This study reports different data than carcass studies” most of the rehab literature is based mainly on live individuals admitted to rehab centres, with a strong bias towards individuals that are still alive. I am not quite sure what “carcass studies” the authors allude to here.

Line 36 – what do the authors mean here by “conservation crisis”?

Introduction

Line 39 – reconsider how the opening sentence begins, as at the moment it is a little difficult to read. What about “Many bird populations throughout North America are declining…”.

Line 43 – could the authors please expand, perhaps with an example, of how a building can be built to be “bird friendly”?

Line 48 – research from the UK has shown a positive relationship between bird-building collisions and increased urbanisation, see: https://doi.org/10.1002/ece3.8856

Line 50 – reconsider the use of “work against birds” here.

Line 64 – insert a comma after “scale”.

Lines 68-69 – the authors claim that most studies use carcass recoveries but only cite a single study here, please add more examples to emphasise the point.

Line 71 – replace “collider” with a more suitable term, here. Either “birds” or “victims” will do.

Lines 71-72 – this sentence could do with some revision, perhaps split into two and emphasise the point that only 7.2% of birds that collided died.

Line 73 – reconsider the word “sharp” here.

Line 74 – replace “to” with “and”.

Lines 90-93 – “, as peer-reviewed articles describing or 93 incorporating rehabilitation data are uncommon [51]” this is simply untrue. There are hundreds of peer-reviewed journal articles that present or use rehabilitation data to answer ecological or conservation research questions. A quick search on Google Scholar will return many, many studies.

Lines 93-95 – again I’m not sure if I agree with this either. The WILD-ONe rehabilitation database is often used in publications. There are also a number of papers published in elsewhere that summarise data from large, collated databases:

https://doi.org/10.1371/journal.pone.0257675

https://doi.org/10.1002/wsb.737

https://doi.org/10.1371/journal.pone.0238805

Line 101 – project?

Lines 107-109 – this is very vague, what do the authors mean by “parts of the avian community”? Please clarify.

Lines 112-113 – I think the authors need to clarify that they are only collecting data from birds that collided with buildings and were admitted to rehabilitation centres. At the moment, this sentence seems to suggest that they will also account for birds that die later on in the wild.

Line 114 – “which theoretically improves their chance at survival.” Be careful with the claims here. The authors do not empirically test survival rates between birds admitted to rehab centres with those that were not admitted, so I don’t think they can make this claim.

Methods

Line 140 – avoid colloquial language throughout, e.g., “there’s”.

Line 145 – “or strong suspicion thereof” what does this mean? This seems very subjective, and I am wondering whether I would be able to follow the authors’ methodology perfectly using that phrase.

Line 146-147 – this sentence can be removed because it has already been made clear that only bird-building collisions records were extracted.

Lines 147-148 – what did the authors do with records that had the disposition as “unknown”? It seems important to me that these are not excluded, but are presented descriptively.

Lines 149-150 – what does this mean? Data were left out because they were too confusing?

Line 160 – correct me if I am wrong but this is the first time I have come across “organisational funding” data? What was this used for? Should be included at the end of the introduction where the authors also outline what they did, etc.

Lines 168-173 – please write in past tense.

Line 170 – “missingness”?

Lines 167-173 – why did the authors attempt to do this? This seems overkill to me, and introduces an entire suite of biases into the data set. What is wrong with modelling with NAs on subset data sets? Even then, this entire subsection is insufficiently described and requires much more extensive methodological details.

Line 176 – R version needed here. Also spell out abbreviations (not for packages/software, etc.).

Line 180 – again, correct me if I am wrong but isn’t a logistic model just a binomial model with a logit link function? If so, it might be best to clarify this for the reader.

Line 181 – I am not sure what a “right-censored survival model” is, could the authors please clarify.

Line 184 – the effects of funding are not included at the end of the introduction (see previous comment). Also, this section needs to be revised/clarified. I am assuming that mass, funding, season, etc., were fitted as explanatory variables? I ask only because this has not been sufficiently explained by the authors here.

Results

Line 211 – I would suggest to avoid starting your results section with a negative, perhaps “We extracted bird-building collisions data from XX..”.

Line 220 – too much unnecessary information here, e.g., (PDFs), some of this could be moved to the supplementary material. Actually, I think the entire first paragraph should be moved to the supplementary material, lead the results section with the *current* second paragraph.

Line 233 onwards – it may be better to present the % in text and the sample size in parentheses, it would help with the flow of the text.

Line 236 – 1/3033 is not 0%, replace with <0.1%.

Line 238 – try to avoid repeating words, e.g., “released”.

Lines 241-242 – output should stay in results section but the description of the statistical tests used to compare differences should be fully explained in the “statistical analyses” subsection of the methods. Furthermore, where are the test statistics and degrees of freedom? This comment also applies throughout.

Line 244 – please write in past tense.

Line 255 – see comment above.

Line 256 – this sentence needs rewording.

Lines 261-268 – this section should be moved up towards the top of the results, perhaps after the sample sizes.

Fig 1 – all figure and table captions should read independently of the main text, i.e., I should know exactly what the authors’ have done without having to read the main text. At present, this caption is lacking information. This comment applies throughout.

Lines 307-308 – models do not “find” anything, alternatively they may “show” a pattern. please reword throughout.

Table 3 – I am confused as to what the 2.5% and 97.5% confidence intervals (?) show here? Aren’t these just lower and upper bounds? In the caption the authors state that 95% Cis were used. There is no mention of any other α-values beside 5% used for the stats. Please clarify.

Discussion

Line 339 – year needed alongside reference.

Lines 352-353 – I think this should be one of your take-home messages from this study, at the moment, it is not highlighted enough.

Lines 370-372 – what about increased light levels during summer which may also increase reflections in glass? Has this ever been considered before?

Line 381 – scientific name required on first mention, please.

Lines 387-388 – one limitation of this study is that it was unable to test intraspecific differences in body mass and the effects on survival, which would be interesting.

Lines 403-405 – could this not just simply be because there are more adults within a population at any one time than juveniles? So the probability of them colliding with buildings is higher anyway?

Lines 413-415 – I really like the idea of exploring the effects of organisational funding, and feel that the authors could emphasize this more especially in the introduction. This is also another one of the study’s important take-home messages. Move towards the top of the Discussion.

Line 423 – this seems like a sensible place to mention large data exercises/databases such as WILD-ONe.

Lines 433-434 – yes, I absolutely agree.

Lines 436-438 – remove this, it comes across as though the authors are dismissing rehabilitation-focused journals here. Actually, having specialised journals is an excellent resource and a good go-to to find studies specifically on wildlife rehabilitation. I would argue that without them, dissemination would be far more challenging than it is now.

Lines 441-455 – references needed throughout this section, there are many studies that mention/discuss biases associated with using rehabilitation centre admissions data. I encourage the authors to ready more widely and with that, cite more widely. Some examples:

https://www.awrc.org.au/uploads/5/8/6/6/5866843/trocini_surveillance.pdf

https://doi.org/10.1002/ece3.8856

Reviewer #3: This paper represents an important perspective for interpreting the true mortality impact of window collisions and is a novel use of wildlife rehabilitation data. After some additional analyses, it could be of interest to a broad audience in the conservation community.

Comments on the analysis:

The analyses were sophisticated, I have not done some of the survival analyses but have experience with MCMC glmm and felt that taxonomic traits and facility parameters required better consideration.

The inclusion of taxonomic traits within the model would increase impact and interest if ‘species’ were included within the model, with an additional correction for non-independence for taxonomic relatedness, perhaps in a covariance matrix (see Barrow et al. 2019). The text states that species or families “were not necessary" (lines 184-186), but I feel that estimating these effects is more instructive than mass or age. There are important ecological traits that predispose particular species to collisions and influence the success of rehabilitation, such as typical species temperament, which can influence the level of care that is possible. Estimating species-specific risk on the taxonomy of collision risk, release probability, and duration of care could help inform wildlife rehabilitation guidelines and provide increased information on species-specific susceptibilities, which could also be implemented in mitigation plans. This was also stated in the discussion (lines 397-399) which I agreed with, and it seems that the data and the analysis would support this.

For the facility, I think overall income would be misleading and should be corrected for facility size, the budget for treatment and standardized per animal treated. Many rehabilitation centres have education and outreach programs, so the expenditure on medical treatment is a more important measure. In my experience, facility funding is critical for increasing survivorship - funding is needed to ensure unexpired medications, oxygen chambers, radiographs and medical consumables for treating shock and avoiding sequelae of traumatic brain injuries. As a message in the paper is the important role of rehabilitation centers and how they are underfunded (line 434), it is critical to provide insight into how funding should be allocated. For example, is it better to fund fewer larger facilities with the capability of engaging more stakeholders, such as veterinarians, whose involvement would help improve outcomes and report quality.

Barrow LN, McNew SM, Mitchell N, Galen SC, Lutz HL, Skeen H, et al. Deeply conserved susceptibility in a multi-host, multi-parasite system. Ecol Lett. 2019;22: 987–998. Pmid:30912262

Comments.

The role of public education in reducing mortality at private residences could be touched upon in the discussion. Very simple window adhesives have been very effective in reducing bird mortality, and it could be beneficial to discuss the products for readers and papers that have tested their efficacy.

Product: https://www.featherfriendly.com/residential

Reviewer #4: The paper is well written, except for some minor copy editing issues. The language is clear and does not emphasize jargon. I did have some difficulty accessing the data spreadsheets. Please see attached document for detailed comments

6. PLOS authors have the option to publish the peer review history of their article (what does this mean?). If published, this will include your full peer review and any attached files.

Reviewer #1: No

Reviewer #2: No

Reviewer #3: No

Reviewer #4: **Yes: **Christine Sheppard

---

## [Author Response · Author response to Decision Letter 0]

11 Jun 2024

[THIS INFORMATION IS PROVIDED IN A BETTER FORMAT AS AN ATTACHED FILE]

Dear Dr. Pukenis Tubelis,

We appreciate the valuable insights and feedback you and the reviewers provided on our manuscript. Please see below for a detailed response to each comment. 

Sincerely,

Ar Kornreich, Dustin Partridge, Mason Youngblood, and Kaitlyn Parkins

Additional Editor Comments:

Please find my review below:

Introduction

Very well written, as occurred with the objectives.

Line 39. Do not indent the first paragraph of a given section. Only the next ones.

Line 46. You have to change to [8 –10]. When you have 3 or more refs in immediate sequence, you have to use a long dash (not hyphen) between the initial and final numbers. Please check this throughout.

Line 48/49. It has to be [11–14].

Line 50. “12” should be the first number within the brackets. Keep always an increasing order.

Line 58. Change to [6, 21–25]. Please follow this format along the manuscript.

These corrections have been made in the manuscript. 

Line 71. I think you have to delete “´s 2022 study”. Just keep the author and the brackets with number.

This has been fixed in the manuscript. 

Line 129. You could add birds and US to the table title.

This has been fixed in the manuscript. 

Line 158. Delete the space between the second bracket and the comma. Delete the comma and space after “location”.

This has been fixed in the manuscript. 

Line 165. I think you have to delete “´s 2014 study”.

This has been fixed in the manuscript. 

Line 244. It is better to change to “varied”. Consider always using past in the Results.

This has been fixed in the manuscript. 

Line 262. Fix the brackets in italics.

This has been fixed in the manuscript. 

Lines 264 to 265. As these numbers after families (n = xx) are found in the Table, there is no need to repeat them here. I suggest deleting them from the text.

This has been fixed in the manuscript. 

Line 270. I think that the figure caption has to be more informative (birds, n states, US…).

This has been fixed in the manuscript.

Line 274. Its title is not much informative. I suggest the same for the figure caption. 

This has been fixed in the manuscript.

Why are the names in italics ? (first column and first line of the table).

This has been fixed in the manuscript. 

Line 313. Table 3. Please check if I´m wright or wrong….should you add something like “Variables” in the cell above that of Mass?

Yes, we have added “Variables” in the cells above “Mass”.

Line 331. Table 4. Explain the bold in some cells. Or change it.

We have removed it, as it is redundant with the asterisks marking the significant effects.

Discussion

I suggest you add Sut-titles for sections of the Discussion. It eases the reding and the finding of specific parts. Not mandatory, but very welcome.

We have added subtitles to the Discussion section.

Line 255. I think you have to delete “2022”.

This has been fixed in the manuscript. 

Lines 423-425. It is a too short paragraph. Can you merge it with other ?

We have expanded on this paragraph in the manuscript to highlight the labor of wildlife rehabbers and the explanatory effect it may have on our findings.

Lines 413-438. Ops! Here you have a full page (19) without the citation of references/papers. Can you please revise this section, and try to add some studies to compare and or support your thoughts/findings. Ah! Now I saw that this problem extends to line 469. Please try to improve this final part of the Discussion. It is too “clean”. Consider an international audience.

We have added citations and more context regarding seasons to increase accessibility with an international audience.

References

Page numbers of articles and chapters have to be separate by a long dash and not hyphen.

You have to add the DOI for all references that have it (mostly articles). Use the format https://doi.org/numbers and letters. Check recent issues if in doubt.

This has been corrected, thank you!

Supplementary Material

Along the text, you used something like “see Supplementary Materials”. You should be more specific, like mentioning the table, figure, etc… Please check the instructions again. Still about this, I could not find the links for them in the pdf that I have to evaluate your submission. I do not know if you provided them or not during the submission. Please check this when you submit the corrected version ok? They will be necessary. Maybe it was not your fault but a problem in the system…

Our deepest apologies for this. The Supplementary Materials were and are accessible through our Data and Code Availability Statement. This has been made clearer in the manuscript. 

5. Review Comments to the Author

Reviewer #1: I consider the results presented in the manuscript to be very beneficial for a better view of the threats to bird species in nature, especially because some of them are still underestimated. The work is clearly written, the literature review and the discussion is of high quality, however, for a better understanding, I suggest some explanation, especially in methodology.

Line 87 - You state that wildlife rehabilitators are licensed by their state - what the conditions are for obtaining a license? I suppose it varies from state to state. Are at least some of the conditions for issuing a license the same or similar, for example, do the conditions for obtaining this license contain certain knowledge (e.g. factors affecting the survival of animals in nature, the basics of treatment, etc.) and their verification, for example by an exam?

We agree that these are all very good questions, but we are unable to address these here because they are out of the scope of this manuscript. A policy review of state policies for granting licenses would be a welcomed addition to the literature and could lead to more cohesive guidelines. 

Line 244 – Why do some seasons include three months and others only two? I assume that in your latitudes the conditions are different from ours, where the year would be divided into four seasons of three months each. On what basis was the division chosen? Can such a division not affect the results of the number of individuals admitted for the given period?

In the global north, season divisions are based on Loss et al. 2014, based on migration and breeding time-frames. More descriptive detail will be added to the manuscript. Because of the seasonal nature of the study area, and the differences in collisions expected between seasons that are regularly studied in the northeast, we believe it is important to divide by biological seasonality rather than by equal numbers of months.

281 – 292 It is not clear how the individual types of injuries differ from each other - for example, how does a concussion differ from a balance issue, as I assume that even with a concussion, balance problems can occur. I would suggest listing the individual categories in the methodology with an example so that it is clear how they differ from each other. Likewise, information on diagnosis is missing - is a veterinarian present and are any special examinations (for example, X-rays) carried out?

Unfortunately, we are bound by the level of detail in the report. Oftentimes, the report simply states injuries as “Head Trauma”, “Concussion”, “Unable to Fly”, etc. With the exception of data from Tri-State Bird Rescue and Research and other very rare exceptions in records from Pennsylvania, in-depth veterinary records were not available to us. 

Line 436 - Is it typical for these rehabilitation centres that they care for only one group of animals (e.g. birds) or are all animals admitted to the rehabilitation centres (including mammals, etc.)? This may also reflect the specialization of these centres and thus better knowledge of the given species.

This varies, but only the minority specialize in only avian species, and most organizations will take in small mammals and/or reptiles as well. There are some rehabbers that primarily or only accept bird-of-prey or songbird species, but the majority or organizations accept most native terrestrial vertebrates. 

—-------------------------------------------------------------------

Reviewer #2: Major comments:

In this study, the authors collate a large data set and explore predictors of bird-building collisions throughout the Northeastern United States. The taxonomic scope of the data set is impressive, spanning some 152 spp., and the authors report significant effects of season and body mass, among others, on aspects of bird-building collisions.

Overall, I thought that the study has merit and could be of interest to a wide audience. However, in its current form I feel that it requires major revision before it can be deemed suitable for publication.

Specifically, as a reviewer I often found myself confused. This often indicates that extra clarification is needed to help the reader fully understand what the authors have done and more importantly, why? This is particularly the case surrounding the statistics employed here. There were some statements which I disagreed with, and I have highlighted these in my list of minor comments below. I encourage the authors to read and cite more widely within the wildlife rehabilitation literature – there are many studies that should be acknowledged here.

Importantly, I was left wondering what the a priori expectations of testing age, body mass, funding, etc., would be given previous knowledge on bird-building collisions. At the end of the introduction, I would expect to find the authors’ expectations evidenced by published research. Inclusion of this, and discussion of each point in sequential order in the Discussion would largely improve the paper.

Given the nature of this study and the findings, I also expect the Discussion section to be full of evidence-informed management actions to prevent bird-building collisions in the future. Currently, there are none. What about decals on windows, habitat management, tinting glass, etc.? This seems like an obvious oversight by the authors and I am sure they will welcome the opportunity to apply their findings in such a way.

As previously mentioned, I believe this study has merit and I hope that the authors receive my comments in a constructive manner, in which they were intended.

Please see point-by-point comments below.

Thank you for your constructive feedback. We have added points to help clarify our intent, added more wildlife rehabilitation literature, and, importantly, added recommendations for how to prevent collisions based on existing literature. 

Minor comments:

Title/abstract – the title refers to “building collisions” but “window collisions” are used throughout the abstract. I would suggest being consistent in use of terminology.

This has been fixed in the manuscript. 

Abstract

Line 19 – replace second “birds” with “those” (the word birds is included three times in one sentence here).

This has been fixed in the manuscript.

Lines 22-24 – there seems to be a sentence missing in your summarised methods here. For example, you state that patterns of survival and release were examined but the next sentence begins with seasonal differences. I suggest adding something along the lines of:

“In addition to this, we examined temporal trends in bird-window collisions and compared differences between species across seasons…”.

Added a note in this sentence to acknowledge we looked at trends across multiple seasons.

Lines 24-26 – split into two separate sentences, it will read better.

This has been fixed in the manuscript. 

Lines 26-27 – this seems a bit obvious, consider omitting to save on word count.

Thank you for the suggestion but while rehabilitators may expect that concussion and head trauma are the most common injury, it is still a result of our study and may be less obvious to readers not as familiar with building collisions. 

Lines 30-31 – this seems to contradict the statement on lines 20-21.

This is precisely one of the reasons we find this study was valuable; it challenges the held assumption that once a patient survives the initial collision, their prognosis is largely going to be fine. Our data suggests that collision victims face serious risk of death for at least 48 hours after the collision occurs. 

Line 32 – “This study reports different data than carcass studies” most of the rehab literature is based mainly on live individuals admitted to rehab centres, with a strong bias towards individuals that are still alive. I am not quite sure what “carcass studies” the authors allude to here.

The majority of field-based studies examining the scale of the impact of building collisions are conducted by examining the number of carcasses found by collision monitors (e.g., Loss et al 2014 & 2015, Parkins et al. 2015, Lao et al 2022). We expand upon this in the introduction and appreciate the comment because it highlights the need for increased collaboration between those in field-based conservation and rehabilitation. We were not aware of many of the valuable rehabilitation papers on this topic until beginning this paper and had solely focused on carcass studies. 

Line 36 – what do the authors mean here by “conservation crisis”?

This has been changed to “this leading threat to wild birds.”

Introduction

Line 39 – reconsider how the opening sentence begins, as at the moment it is a little difficult to read. What about “Many bird populations throughout North America are declining…”.

This has been fixed in the manuscript. 

Line 43 – could the authors please expand, perhaps with an example, of how a building can be built to be “bird friendly”?

This is an important suggestion for protecting birds and we have added this to the conclusion. 

Line 48 – research from the UK has shown a positive relationship between bird-building collisions and increased urbanisation, see: https://doi.org/10.1002/ece3.8856

Thank you for this! That study is now cited in that portion of the manuscript, as citation 11. 

Line 50 – reconsider the use of “work against birds” here.

This has been changed “work against” to “endanger”

Line 64 – insert a comma after “scale”.

This has been fixed in the manuscript. 

Lines 68-69 – the authors claim that most studies use carcass recoveries but only cite a single study here, please add more examples to emphasise the point.

These have been added to the manuscript. 

Line 71 – replace “collider” with a more suitable term, here. Either “birds” or “victims” will do.

This has been fixed in the manuscript. 

Lines 71-72 – this sentence could do with some revision, perhaps split into two and emphasise the point that only 7.2% of birds that collided died.

This has been fixed in the manuscript. 

Line 73 – reconsider the word “sharp” here.

This has been changed to “drastic” in the manuscript.

Line 74 – replace “to” with “and”.

This has been fixed in the manuscript. 

Lines 90-93 – “, as peer-reviewed articles describing or 93 incorporating rehabilitation data are uncommon [51]” this is simply untrue. There are hundreds of peer-reviewed journal articles that present or use rehabilitation data to answer ecological or conservation research questions. A quick search on Google Scholar will return many, many studies

This has been fixed in the manuscript– we have removed this line. 

Lines 93-95 – again I’m not sure if I agree with this either. The WILD-ONe rehabilitation database is often used in publications. There are also a number of papers published in elsewhere that summarise data from large, collated databases

Thank you for alerting us to this resource, we have added citations including the usage of these databases, but we still feel as though rehabilitation-based resources are underutilized in conservation research, and that communication between rehabilitators, conservation scientists, and other stakeholders could be improved. 

Line 101 – project? 

This has been changed to “study” in t

---

## [Editor Report · Decision Letter 1]

16 Jun 2024

Rehabilitation outcomes of bird-building collision victims in the Northeastern United States

PONE-D-24-13495R1

Dear Dr. Ariella Kornreich and Dr Dustin Partridge,

We’re pleased to inform you that your manuscript has been judged scientifically suitable for publication and will be formally accepted for publication once it meets all outstanding technical requirements.

Kind regards,

Dárius Pukenis Tubelis, Ph.D.

Academic Editor

PLOS ONE

Additional Editor Comments:

Dear Dr Ariella Kornreich and Dr Dustin Partridge,

Thank you for submitting the corrected version of your submission about bird collisions in the Northeastern United States (PONE-D-24-13495R1).

I carefully read this corrected version and the responses that you provided to the reviewers´ comments and suggestions.

I consider that the responses provided by your research team are quite appropriate and convincing.

Also, I agree with the changes that you have done on the manuscript based on the five reviews. I noted that you followed most suggestions.

With this, the quality of your submission is even higher.

Therefore, I consider that your study reaches the standard expected for publication in PLOS ONE. It is in adherence to the PLOS ONE publication criteria.

Thus, I suggest that this submission be accepted in definitive for publication in PLOS ONE.

During my last reading, I found a set of minor errors that can be easily fixed by you along the next days.

They are listed below. Please check them.

PLOS ONE people might contact you soon for final actions.

Thank you for considering PLOS ONE as home of your research.

Your study will be a great contribution to the journal.

Please feel free to contact me if you have any doubts regarding this submission.

Dr Dárius P. Tubelis

PLOS ONE Editor

**Last things to fix:**

Line 88. Delete "a" after "47".

Line 130. I think the correct is "Table 1." (instead of ":"). Also, add a dot point after "States".

Line 162. Would it be "4999"? Change, if so.

Line 169. I think you have to delete ")"; see the end.

Line 169. About "averaged low and high values.". It sounds stange. Is it correct ? If yes, please forgive me. If there is something missing, please fix.

Lines175-177. This does not appear to be part of "Acquiring" (the first section)..... can you move to other section below? It sounds strange here. Or make it an independent section....

Line 234. Fix one of the two commas.

Line 246. Add a comma for "1644".

Line 257. The left bracket for "188" is in italics.

Line 264. I think the correct is "Fig 1."

Line 270. The same. Replace ":" by "." after Table.

Line 278. It should be "3,033".

Line 278. I think you could delete "of cases" after "39.5%".

Line 278. Similarly, you might delete "of cases of" after "32.1%". Note two "of".

Line 279-280. Also delete "of cases of" after "28.4%".

Line 282. Maybe you can delete "cases" after "127".

Line 283. There is something wrong with "arrival cases". Or not ?

Lines 283-284. At the end, you wrote "case cases"....it sounds wrong.

Line 292. "1063". You were using a comma for thousands.

Line 297. Do not use "Summer" with capital to keep a pattern.

Line 303. I think you have to replace "X2" by the correct symbol, if you are referring to Chi-square. Please check the values of these two lines again, to make sure to report them correctly.

The same for lines 307 and 308.

Line 304. Do not start a sentence with a number. Maybe you can add " A total of".

Line 310. You have to add a comma to "1217" to keep your pattern.

Line 342. Use "Table 2." Also, add a dot point after the last word of the title.

Lines 348-349. The same for the figure caption.

Lines 364-365. The same.

Line 376. I suspect you can delete "2014" as you cited "7" ahead.

Line 382. Add a space after "buildings".

Line 436. Replace "&" by "and".

Line 452. Add a space before "71".

References

They appear to be well formatted.

But when you have more than 6 authors, you have to cite the first 6 and then ", et al." Please see instructions or recent papers.

Good work!

Dárius

---

## [Editor Report · Acceptance letter]

11 Jul 2024

PONE-D-24-13495R1 

PLOS ONE

Dear Dr. Kornreich, 

I'm pleased to inform you that your manuscript has been deemed suitable for publication in PLOS ONE. Congratulations! Your manuscript is now being handed over to our production team.

Kind regards, 

on behalf of

Dr. Dárius Pukenis Tubelis 

Academic Editor

PLOS ONE